# Unraveling exciton–phonon coupling in individual FAPbI$_3$ nanocrystals emitting near-infrared single photons

Ming Fu[1,2], Philippe Tamarat[1,2], Jean-Baptiste Trebbia[1,2], Maryna I. Bodnarchuk[3], Maksym V. Kovalenko [3,4], Jacky Even [5] & Brahim Lounis[1,2]

Formamidinium lead iodide (FAPbI$_3$) exhibits the narrowest bandgap energy among lead halide perovskites, thus playing a pivotal role for the development of photovoltaics and near-infrared classical or quantum light sources. Here, we unveil the fundamental properties of FAPbI$_3$ by spectroscopic investigations of nanocrystals of this material at the single-particle level. We show that these nanocrystals deliver near-infrared single photons suitable for quantum communication. Moreover, the low temperature photoluminescence spectra of FAPbI$_3$ nanocrystals reveal the optical phonon modes responsible for the emission line broadening with temperature and a vanishing exciton–acoustic phonon interaction in these soft materials. The photoluminescence decays are governed by thermal mixing between fine structure states, with a two-optical phonon Raman scattering process. These results point to a strong Frölich interaction and to a phonon glass character that weakens the interactions of charge carriers with acoustic phonons and thus impacts their relaxation and mobility in these perovskites.

[1] Université de Bordeaux, LP2N, Talence F-33405, France. [2] Institut d'Optique and CNRS, LP2N, Talence F-33405, France. [3] Laboratory for Thin Films and Photovoltaics, Empa-Swiss Federal Laboratories for Materials Science and Technology, Dübendorf CH-8600, Switzerland. [4] Institute of Inorganic Chemistry, Department of Chemistry and Applied Biosciences, ETH Zürich, Zürich CH-8093, Switzerland. [5] Univ Rennes, INSA Rennes, CNRS, Institut FOTON - UMR 6082, Rennes F-35000, France. Correspondence and requests for materials should be addressed to B.L. (email: brahim.lounis@u-bordeaux.fr)

Metal halide perovskites, in particular hybrid organic–inorganic lead halides $APbX_3$, where A is an organic cation (methylammonium, MA, or formamidinium, FA) and X is halide (Cl, Br, I), have attracted a vast interest over the past few years[1]. Their outstanding optical and electronic properties, together with facile and cost-effective production, make these materials useful not only for photovoltaic applications[2–5], but also for light-emitting devices[6–9]. Recent advances in the colloidal synthesis of strongly emitting perovskite nanocrystals (NCs) open up new possibilities for the fabrication of tunable light sources based on the composition and quantum size effect tuning, such as light-emitting diodes and lasers, and for the exploration of their potential use as quantum light sources[10–16]. Previous works demonstrating single-photon emission of perovskites are mainly focused on fully inorganic cesium lead halide perovskites NCs ($CsPbX_3$), which emit in the visible wavelength region[17–19]. It was also shown recently that hybrid organic–inorganic lead halide $FAPbBr_3$ NCs behave as efficient single-photon sources in the visible domain[20]. Yet, the potential of perovskite NCs as quantum light sources in the near-infrared remains unexplored, in spite of important benefits concerning the tunability of the emission lines to the maximal sensitivity of silicon avalanche photodiodes and their suitability for propagation in telecommunication fibers. So far, red-emissive $CsPbI_3$ NCs have been shown to suffer from a delayed phase transformation into a non-luminescent wide-band-gap 1D polymorph, and $MAPbI_3$ exhibits very limited chemical durability and eventually decompose to $PbI_2$. This so-called "perovskite red wall" has only recently been overcome with the development of a colloidal synthesis method for obtaining phase-stable FA lead iodide ($FAPbI_3$) and $FA_{1-x}Cs_xPbI_3$ NCs with a bright photoluminescence (PL) expanding to 800 nm in the near-infrared[21].

Bright light sources that deliver on demand single indistinguishable photons are needed for a variety of quantum information processing schemes[22]. Those based on solid-state quantum emitters require cryogenic temperatures to reduce phonon dephasing of the transition dipole and to obtain a sharp zero-phonon emission line (ZPL). Indeed, indistinguishable photons stem from the ZPL when its linewidth reaches its lower bound set by the lifetime of the emitting state. Therefore, quantum emitters that exhibit a weak coupling between the exciton and phonons are appealing for the development of those applications. Besides governing the emission line broadening of hybrid perovskites, phonon scattering is among the factors setting a fundamental intrinsic limit to the mobility of charge carriers in these materials[23]. The study of electron–phonon coupling in perovskite NCs is thus important for photovoltaic applications. Yet, the strong inhomogeneous line broadening, inherent to PL spectra of bulk samples[24,25] or ensembles of NCs[26], obscure the spectral features of phonons and lead to disparities in the extracted values of phonon energies and coupling strengths.

Halide perovskite semiconductors exhibit fundamental differences with classical semiconductors with zinc-blende or würtzite crystal structures. The spin-orbit coupling effect, which is very strong for lead-based compounds, is indeed present in the conduction band rather than in the valence band, leading to doubly spin-degenerated band-edge electronic states[27]. Moreover, the selection rules for carrier–phonon interactions are specific and similar for both electrons and holes[28,29]. In particular, polar acoustic phonon mechanisms related to piezoelectricity and non-polar optical phonon mechanisms related to deformation potentials vanish in the cubic Pm-3m reference phase. Carrier–phonon interactions are thus related to acoustic phonons via deformation potentials and to polar optical phonons. The Fröhlich coupling is expected to dominate in these strongly ionic materials at room temperature[30]. Furthermore, inorganic halide perovskites display three triply degenerated optical active lattice modes instead of one in classical semiconductors[31]. These modes are expected to undergo longitudinal optical (LO)–transverse optical (TO) phonon splittings, which strongly depend on the mode polarity. For example, in inorganic $CsPbCl_3$ perovskites, the highest-energy optical mode exhibits a significant LO–TO splitting (25 meV–15 meV), which is almost entirely at the origin of the difference between the static and optical dielectric constants[32]. In the well-known iodine-based hybrid halide perovskite $MAPbI_3$, the same difference is explained at room temperature by a LO–TO splitting at lower frequencies (5 meV–4 meV)[33]. Recent Terahertz emission spectroscopy experiments indeed show that the photogenerated carriers drive an ultrafast coherent lattice response around 5 meV with a somewhat broader feature at about 11.6 meV[34]. There is scarce information about the optical phonons in $FAPbI_3$. An optical mode at ~ 17 meV is observed at room temperature in the bulk α-phase by Raman spectroscopy[35,36], and terahertz emission spectroscopy shows two coherent emission signatures at about 5 meV and 10 meV in $FAPbI_3$-based alloys[34].

Lead halide perovskites are structurally soft semiconductors having their whole acoustic vibrational density of states located at low energies (in the meV range)[37]. Recent inelastic neutron-scattering measurements[38] indicate that $FAPbI_3$ is an ultra-soft halide perovskite material with a very low elastic stiffness. These measurements are consistent with the prediction of a strong acoustic-optical phonon up-conversion and "hot-phonon bottle-neck" in hybrid organic–inorganic lead halide perovskites[39], especially in $FAPbI_3$. Indeed, the blocking of acoustic phonon propagation, as well as the presence of organic cations introducing overlapping acoustic and optical phonon branches, is expected to facilitate the up-transition (depopulation) of low-energy acoustic modes to optical modes. An alternative explanation on the role of low-energy acoustic vibrational density of states is connected to a glassy state, where the strong lattice anharmonicity impedes the scattering of electron by acoustic phonons. This latter phenomenon is also expected to dominate the processes of band-edge electron coupling to acoustic phonons and thus charge carrier transport[40]. These processes prolonging the cooling period of hot charge carriers and affecting charge transport are promising for the next-generation photovoltaics devices, and motivate further investigations of the role of acoustic phonons in the electron–phonon interactions in $FAPbI_3$ materials.

In this article, we present a spectroscopic study of these near-infrared perovskites at the single NC level and demonstrate the strong photon antibunching of their emission at room temperature. We also investigate their temperature-dependent PL spectra and PL decay at cryogenic temperatures. We find that the homogeneous broadening of the PL spectrum with increasing temperature is negligible up to 30 K, revealing a weak exciton–acoustic phonon coupling in these materials. We demonstrate that the emission line broadening is governed by the coupling between excitons and optical phonon modes whose spectral signatures are identified in the low temperature PL spectra. Moreover, the temperature dependence of the PL decay unveils a thermal mixing mechanism between the lowest two levels of the band-edge exciton fine structure, which is based on a two-phonon Raman process with optical phonons.

## Results

**Room temperature properties of $FAPbI_3$ NCs.** $FAPbI_3$ NCs with a schematic crystal structure presented in Fig. 1a were synthesized by a wet-chemical method reported previously[21], refined for improved size and shape uniformity (see Methods for

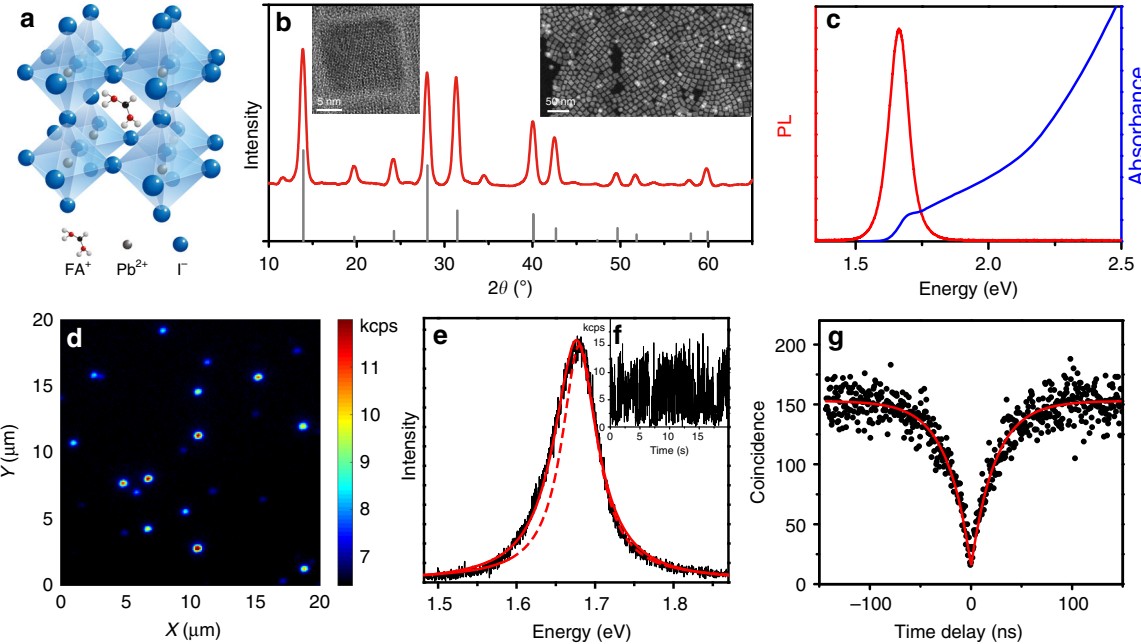

**Fig. 1** Room temperature properties of FAPbI$_3$ perovskite NCs. **a** Scheme of the cubic perovskite crystal structure of FAPbI$_3$. **b** Powder X-ray diffraction pattern for FAPbI$_3$ NCs (red line), along with the simulated pattern of bulk FAPbI$_3$ from the ICSD database (card number 250736, gray). Right image: High-angle annular dark-field scanning transmission electron microscopy image (HAADF-STEM) from an ensemble of FAPbI$_3$ NCs. Left image: High-resolution transmission electron microscopy image (HR-TEM) of a single FAPbI$_3$ NC. **c** Absorption and PL spectra of FAPbI$_3$ NCs in a toluene solution. **d** Wide-field PL image of single FAPbI$_3$ NCs dispersed in poly(methyl methacrylate) at room temperature, measured using a 488 nm continuous-wave excitation. The color bar marks the PL intensity measured. **e** PL spectrum of a FAPbI$_3$ NC recorded in a confocal mode over 5 s with an excitation intensity of 6 kW cm$^{-2}$. The dashed curve shows a Lorentzian profile, which reproduces well the high-energy wing of the emission line. It points to an asymmetry of the PL spectrum. The spectrum is reproduced with a double-Lorentzian profile (plain red curve), comprising the zero-phonon line and an optical phonon sideband red-shifted by 12 meV. **f** Time trace of the PL intensity of a FAPbI$_3$ NC, measured under an excitation intensity of 2 kW cm$^{-2}$ and with a bin time of 10 ms. **g** Histogram of time delays between consecutive photon pairs detected from the PL of a FAPbI$_3$ NC under an excitation intensity of 2 kW cm$^{-2}$

details). High-resolution transmission electron microscopy (HR-TEM) images and high-angle annular dark-field scanning transmission electron microscopy images (HAADF-STEM) reveal that these NCs have cuboid shapes with average sizes of 10–15 nm (Fig. 1b), i.e., 2–3 times larger than the exciton Bohr radius in FAPbI$_3$[41]. As shown in Fig. 1b, powder X-ray diffraction measurements identify a cubic perovskite crystal structure at room temperature. The optical absorption spectrum of the FAPbI$_3$ NCs dissolved in toluene extends from UV to the near-infrared absorption, with the absorption edge at ~1.70 eV (Fig. 1c). Under visible light excitation, the NCs exhibit a bright near-infrared PL centered at 1.66 eV, with a high color purity (full width at half-maximum, FWHM = 88 meV) and a high PL quantum yield over 70%[21].

**Producing single near-infrared photons with single FAPbI$_3$ NCs.** We use single NC spectroscopy to get rid of inhomogeneities due to size and morphological variations among these nanostructures and thus explore their intrinsic optical properties, as well as to probe limiting carrier scattering processes at low temperature. An inverted fluorescence microscope equipped with a 1.45 numerical aperture oil-immersion objective is used to investigate the room temperature properties of single FAPbI$_3$ NCs deposited by spin-coating on a clean glass coverslip. Figure 1d shows a representative wide-field image of single FAPbI$_3$ NCs dispersed in a poly(methyl methacrylate) matrix and excited at 488 nm. The disparity between PL intensities among these NCs is attributed to the dispersion of their quantum yields around the ensemble-averaged value[42]. A typical PL spectrum of a single NC

is shown in Fig. 1e. Its linewidth of FWHM ~ 70 meV is almost as large as that of the ensemble PL spectrum of Fig. 1c. This small inhomogeneous broadening in spite of the NC size dispersion reflects the weak quantum confinement effect in these NCs. The asymmetry displayed in the PL spectral profile is the hallmark of an optical phonon sideband, although a residual contribution from a low emissive state to the red shoulder cannot be ruled out. As shown in Fig. 1f, individual NCs exhibit a characteristic PL blinking behavior, which is likely due to random charging and discharging processes of the NC driven by photoionization, with a domination of Auger recombination in the charged exciton decay[17,18]. Although negligible in bulk semiconductors, the Auger effect is strong in NCs due to the enhanced Coulomb interaction between charge carriers and the reduced kinematic restriction on momentum conservation[43]. In spite of the weak quantum confinement in these FAPbI$_3$ NCs, the Auger processes also provide an efficient channel for nonradiative recombination of multiple excitons, which prevents the emission of multiple photons. Indeed, a remarkable feature of the PL from these single NCs is the strong photon antibunching, which manifests as a dip in its normalized intensity autocorrelation function $g^{(2)}(\tau)$ at zero delay time ($\tau = 0$), as exemplified in Fig. 1g (See other examples in Supplementary Fig. 1). We find $g^{(2)}(0) \sim 0.1$, close to the ideal signature $g^{(2)}(0) = 0$ of a pure single-photon stream. As single NCs are detected with a high signal-to-background ratio, the correlation of the PL signal with background events only contributes as ~0.02 to $g^{(2)}(\tau)$. The main contribution to $g^{(2)}(0)$ is therefore due to residual biexciton radiative recombinations. Similar values of $g^{(2)}(0)$ were reported in the case of individual perovskite NCs of CsPbX$_3$[17,18] or FAPbBr$_3$[20] emitting in the visible domain. The

recovery of the coincidence signal with time is well fitted by a monoexponential rise function, which leads to an exciton lifetime of 25 ns at room temperature for this NC.

**Low temperature PL spectra of single NCs.** To gain a deeper understanding of the optical properties of FAPbI$_3$ NCs, we have investigated the PL spectrum of more than a hundred of single NCs at cryogenic temperatures. At the lowest temperatures (~3.6 K), thermal dephasing is reduced and the PL spectra of single NCs mainly present a sharp and intense line attributed to the exciton recombination ZPL, as exemplified in Fig. 2. The ZPL undergoes a small spectral diffusion with an excursion contained

within a few meV (see Supplementary Fig. 2). This behavior stands apart from that of fully inorganic perovskite NCs, which display a remarkable spectral stability in the low temperature PL spectra[19,44,45]. A possible origin of the spectral diffusion can be explained as follows. Contrary to cations in fully inorganic perovskites, organic cations in hybrid lead halide perovskites are known to have electric dipole and quadrupole moments[46]. Photogenerated charge carriers cause the neighboring cations to reorient[47] and locally align their dipoles to form ferroelectric domains[48–50] and maximize the screening effect. This results in a change of the electrostatic potential in the NC, leading to a shift of the transition energy of the exciton. It is consistent with the recently reported locally disordered low temperature phase of

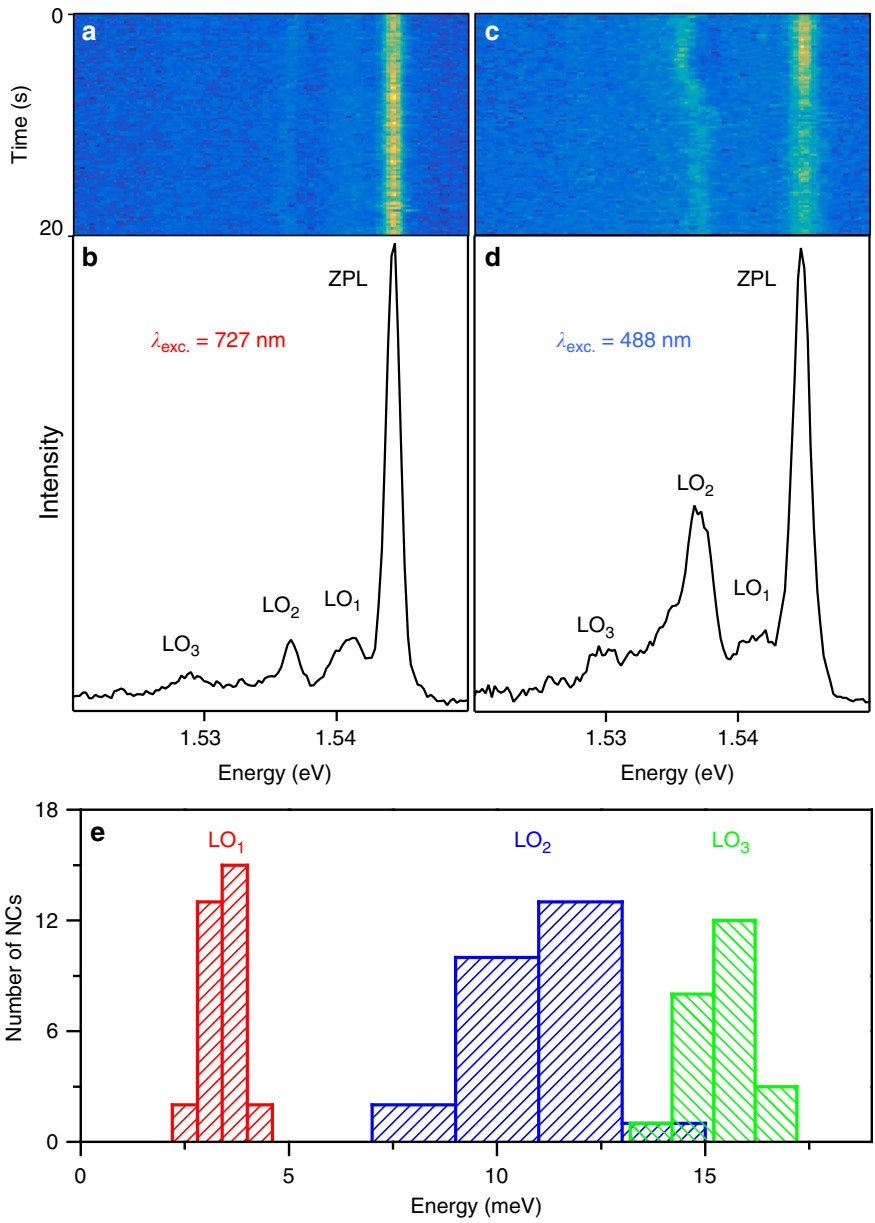

**Fig. 2** PL spectrum of a single FAPbI$_3$ NC at 3.6 K. **a** Spectral trajectory of a single NC at 3.6 K, built with 100 consecutive PL spectra, each recorded over 0.2 s at an excitation wavelength of 727 nm. The resolution of the spectrograph is 0.8 meV (using a grating of 300 lines/mm). In this trajectory, the spectra were shifted in order to eliminate the ZPL spectral diffusion. **b** PL spectrum of the same NC, built from the sum of all spectra of **a**. The energies of the LO$_1$ LO$_2$, LO$_3$ optical phonon modes are 3.2 meV, 7.8 meV, 15.4 meV, respectively, and the ZPL linewidth is 1.2 meV FWHM. **c** Spectral trajectory of the same NC excited at 488 nm, with corrections from spectral diffusion of the ZPL as in **a**. **d** PL spectrum built from the sum of all spectra of **c**. The ZPL linewidth is 1.5 meV FWHM. **e** Histogram of LO$_1$ LO$_2$, LO$_3$ phonon energies

FAPbI$_3$[51]. During the acquisition of the PL spectra, random configurations of cation dipole organizations (domain size and orientation) will be sampled and lead to spectral diffusion of the exciton recombination line. The line shape is thus essentially set by the distribution of emission line frequencies sampled during the integration time of the spectra. The random process at the origin of spectral diffusion leads to a Gaussian-shaped emission line. The sharpest lines are obtained when reducing the integration time, as well as the excess energy between the excitation photons and the emission photons. Linewidths as narrow as 0.8 meV can be obtained when recording the spectra over a very short integration time (0.2 s) and by using an excitation wavelength in the near-infrared (~730 nm) (see Fig. 2 and Supplementary Fig. 2). Yet, the linewidths are not limited by the resolution of the spectrograph and may hide an unresolved band-edge exciton fine structure, which has been observed in inorganic lead halide perovskite NCs[44,45,52–55].

As exemplified in Fig. 2, the low temperature PL spectra display three phonon replicas that are red-shifted by ~3–4 meV, ~10–12 meV and ~14–16 meV (see the histograms of Fig. 2e and Supplementary Figs 2-7, 9 for further examples of PL spectra). These sidebands are attributed to LO phonon modes and labeled LO$_1$, LO$_2$, LO$_3$, respectively. The LO$_1$ and LO$_2$ modes lie in the same energy range as the coherent lattice emission signatures observed by terahertz spectroscopy[34]. Their Huang-Rhys factors are one order of magnitude larger than those measured in CsPbBr$_3$[44] and CdSe[56] NCs, indicating a stronger exciton–phonon coupling in FAPbI$_3$ NCs. Interestingly, Fig. 2 shows evidence for a drastic increase in the LO$_2$ sideband intensity when changing the excitation wavelength from $\lambda_{exc}$ = 727 nm (quasi-resonant) to $\lambda_{exc}$ = 488 nm (far above resonance). Moreover, its relative spectral position with respect to the ZPL

undergoes significant fluctuations over time under blue excitation (Fig. 2c). On the basis of theoretical predictions combined with near-infrared spectroscopic measurements on MAPbI$_3$ thin films at low temperature[57], the LO$_{1–3}$ bands identified in Fig. 2 can be assigned to three bundles of finely separated low-energy lattice modes. These modes correspond to stretching and bending vibrations of the PbI$_3$ network and rigid-body motion of the FA cation, whose character is affected by their mutual couplings[57]. We attribute the temporal fluctuations of the phonon sideband position to fluctuations of the halide structure subsequent to the activation of cation rotation with high-energy photons, as evidenced in MAPbI$_3$ by time-resolved electron scattering techniques[58]. We suggest that mixed organic–inorganic electronic states are excited in FAPbI$_3$ NCs at high energy, as in the case of MAPbI$_3$[59]. After fast relaxation of charge carriers to the band-edge, structural deformations are induced with preferential couplings to specific lattice modes. All along the NC spectral trajectory, various relaxation configurations and therefore LO vibrations are probed over the excitation cycles.

**Evolution of the PL spectra with temperature.** In the following, we investigate how the emission properties of single NCs evolve with temperature. Figure 3 displays the PL spectra of a single NC recorded from liquid helium temperature up to 150 K (other examples of temperature-dependent PL spectra are shown in Supplementary Figs 4-7). The emission lines experience a blue shift of ~25 meV when increasing the temperature up to 90 K. Such a blue shift is counter intuitive with respect to empirical Varshni[60], Pässler[61] and Bose-Einstein[62] models used to describe the usual bandgap shortening with increasing temperature observed for conventional semiconductors. These models are no

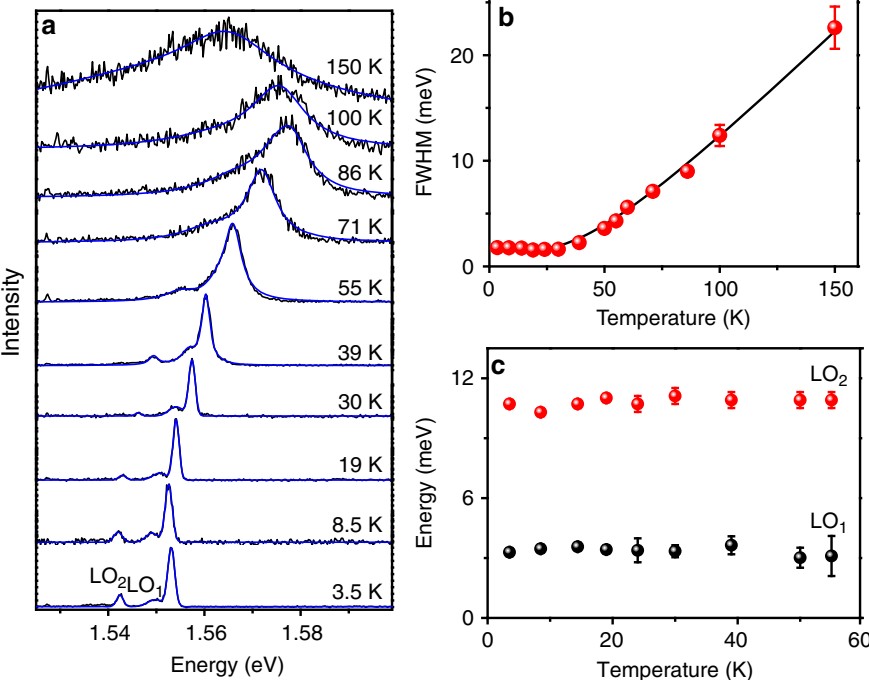

**Fig. 3** Temperature dependence of the PL spectrum of a single FAPbI$_3$ NC. **a** Evolution of the PL spectrum of a FAPbI$_3$ NC with temperature. Each spectrum was recorded with an excitation intensity of 700 W cm$^{-2}$. The spectra are fitted (blue lines) with Gaussian profiles up to 30 K and with Lorentzian profiles above 30 K (see text). **b** ZPL linewidth (FWHM) of the same NC as a function of temperature. The black line is a fitting curve taking into account only the Fröhlich coupling between the exciton and an effective LO phonon mode with energy $E_{LO}$ = (10.7 ± 0.8) meV. The low temperature linewidth is $\Gamma_0$ = 1.5 meV. The contribution of acoustic phonon to the line broadening is negligible as $\sigma_{Ac}$ < 5 µeV K$^{-1}$. **c** LO$_1$ and LO$_2$ phonon energies as a function of temperature from 3.5 K to 55 K, deduced from the fits in **a**. The error bars displayed in **b** and **c** represent the fitting errors of the parameters deduced from **a**

longer suitable for such materials, due to the complex interplay between the electron–phonon renormalization and the thermal expansion, which have opposite effects on the bandgap energy upon a variation of temperature. In fact, this atypical bandgap widening with increasing temperature was previously observed in bulk hybrid lead halide perovskites[25,63,64] and assigned to a stabilization of the valence band maximum as the lattice expands with temperature[63]. Strikingly, when the temperature is raised above 100 K, the temperature drift of the NCs PL spectrum changes sign and turns to a red shift. This behavior is reproduced for all other single FAPbI$_3$ NCs (see other examples in the Supplementary Figs 4-7). These evolutions remind complex spectral trajectories of bulk hybrid lead halide perovskites as a function of temperature, which were associated with phase transitions found in these relatively soft materials[24,25,63,65,66]. It is thus a remarkable feature that a smooth phase transition over temperature can occur at the scale of a single NC. In the case of FAPbI$_3$ films, a spectral evolution similar to that obtained with single NCs has been observed and the nonmonotonic behavior found around 130 K attributed to a structural phase transition between trigonal phases[24,67]. The existence of a low temperature monoclinic phase was postulated from the inspection of X-ray diffraction data at ~ 100 K[68], but the structure was not determined. No evidence of a monoclinic to trigonal transition could be brought with the PL spectra of FAPbI$_3$ bulk samples or NCs. The possible existence of a monoclinic phase at low temperature requires additional experimental studies.

Compared with spectroscopic studies of FAPbI$_3$ bulk samples or ensemble of NCs where the exciton recombination lines are strongly affected by inhomogeneous broadening[24–26], single NC studies allow an accurate determination of acoustic and optical phonons contributions to the thermal broadening of the homogeneous line. Above 30 K, the homogeneous line broadening by thermal dephasing prevails over the spectral excursions caused by spectral diffusion. The emission line thus acquires a Lorentzian shape that is characteristic of a homogeneous line at higher temperatures, as exemplified in Fig. 3a. This broadening with temperature $T$ is reproduced using the expression $\Gamma(T) = \Gamma_0 + \sigma_{Ac}T + \Gamma_{LO}/[\exp(E_{LO}/k_BT)-1]$[69,70], where $\Gamma_0$ is the zero-temperature ZPL linewidth, which is temperature independent and mainly set by spectral diffusion, $\sigma_{Ac}$ and $\Gamma_{LO}$ are, respectively, the exciton–acoustic phonon and the exciton–optical phonon coupling coefficients, and $E_{LO}$ is the optical phonon energy. The contribution of optical phonons to the line broadening is indeed proportional to their Bose–Einstein occupation number, whereas for acoustic phonons a linear dependence on temperature is generally assumed. We find that coupling to a single optical phonon mode is enough to reproduce the temperature evolution of the linewidth, whereas more than one polar optical modes are present in halide perovskites. We thus assume that the $E_{LO}$ parameter is an effective parameter representing the carrier–LO phonon interaction. In the example of Fig. 3b, the fit is performed with $\Gamma_0 = 1.5$ meV, $\sigma_{Ac} = 0$, $\gamma_{LO} = 27$ meV and $E_{LO} = 10.7$ meV. The values of $E_{LO}$ and $\gamma_{LO}$ are in agreement with previous measurements on bulk FAPbI$_3$[25]. However, this value of $E_{LO}$ differs from that deduced from other measurements on bulk FAPbI$_3$ and FAPbI$_3$ NC ensembles[24,26], providing values $E_{LO}$ ~18 meV. Our single NC studies are self-consistent as the LO-phonon modes participating to the homogeneous broadening are directly observed in the low temperature PL spectra, while buried in the inhomogeneously broadened spectrum of the bulk material or NC ensembles (larger by one order of magnitude). It is worth noting that the energies of LO$_1$ and LO$_2$ phonon sidebands are very stable with temperature (see Fig. 3c). This points to an absence of phonon softening effect, in contrast with observations on the inorganic perovskite CsPbCl$_3$[32].

Strikingly, the acoustic phonon contribution, which should dominate at low temperature (below 30 K), is found negligible for all the studied NCs. We find an upper bound for $\sigma_{Ac}$ of ~5 μeV K$^{-1}$, a value that is more than one order of magnitude weaker than that estimated from measurements in bulk FAPbI$_3$[24,25] and comparable to that of inorganic perovskite NCs[54,71]. Therefore, the acoustic phonon contribution to the linewidth broadening at room temperature is negligible, the major one being that of intrinsic Fröhlich interaction between the exciton and LO phonon modes. The acoustic phonon coupling coefficient can be theoretically estimated using the expression[72,73]:

$$\sigma_{Ac} = \frac{M^2 D^2 k_B}{\pi \hbar^3 \rho v_s} \tag{1}$$

where $D$ is the deformation potential, $M$ the electron and hole total mass, $k_B$ the Boltzmann constant, $\rho$ the mass density and $v_s$ the sound velocity. Taking $M = 0.4m_e$, an average sound velocity $v_s = 1270$ m s$^{-1}$ [38,74], $D = 5$ eV[75], $\rho = 4.1$ g.cm$^{-3}$ [68], we find $\sigma_{Ac} = 0.4$ μeV K$^{-1}$. Due to the lack of information in this perovskite, the exciton polaritonic effects are not taken into account in this simple estimation. They may lead to an enhancement of $\sigma_{Ac}$ by a factor of up to 5[73]. Nevertheless, from this estimation and our measurements, we infer that the larger $\sigma_{Ac}$ values measured in bulk FAPbI$_3$[24,25] are merely related to extrinsic effects rather than intrinsic electron–phonon interactions.

**Evolution of the PL decay with temperature**. In order to explore the exciton recombination dynamics and relaxation rates within the band-edge exciton fine structure, we investigated the temperature dependence of the PL decay of single FAPbI$_3$ NCs. Figure 4a shows a representative evolution of the PL decay of a NC with temperature. At $T = 3.5$ K, the PL decay exhibits a monoexponential behavior with a lifetime of 1.5 ns. With a slight increase in temperature by a few Kelvin, a long decay component shows up in the 50 ns range. With a further increase in temperature, the PL decay displays a clear biexponential character with a long decay component that gains weight and shortens with temperature, whereas the short component shortens and loses weight until it vanishes above 70 K. Moreover, under magnetic fields, the PL decay shows a shortening of its long-lived component with an increase of its weight (See Supplementary Fig. 8). These characteristic behaviors are obvious signatures of thermal mixing and magnetic coupling between bright and dark fine structure sublevels, and are similar to that previously reported in CdSe NCs[76–78]. For FAPbI$_3$ NCs, no spectroscopic study of the band-edge exciton fine structure is reported. Nevertheless, this fine structure should be weakly influenced by the nature of the cation, and thus be similar to that of CsPbI$_3$. It comprises a bright triplet, whose degeneracy may be lifted by crystal field depending on the crystal structure and a dark single state[44,53]. Fine structure splitting measurements, based on PL spectra of single CsPbI$_3$ NCs, led to a statistical distribution with an average splitting of ~300 μeV and an upper value of ~500 μeV[45]. To unveil a dark state emission line via magnetic brightening, we recorded PL spectra of individual FAPbI$_3$ NCs under external magnetic fields up to 7 T. We did not observe any emerging peak but a slight emission line broadening (Supplementary Fig. 9). This suggests that the dark exciton level lies close to a bright level, with a bright-dark energy separation within a linewidth, i.e., in the range of few hundred μeV.

The temperature dependence of the PL decay can be reproduced with a simple model assuming a thermal mixing between two fine structure energy levels: one related to the dark state and the other to the bright triplet manifold. We thus use a three-level system comprising the zero-exciton ground level |G⟩

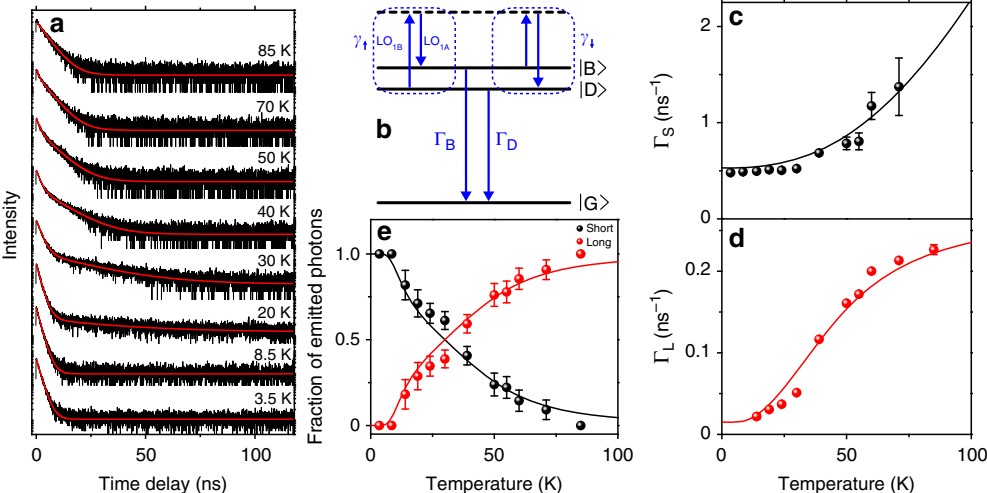

**Fig. 4** Evolution of the PL decay of a single FAPbI$_3$ NC with temperature. **a** PL decays of the same FAPbI$_3$ NC at various temperatures, under pulsed excitation at 488 nm with an average intensity of 30 W cm$^{-2}$. A log-log scale representation is displayed in Supplementary Fig. 12. **b** The three-states model. $|G\rangle$ is the zero-exciton state, $|D\rangle$ and $|B\rangle$ are the lowest dark and bright excitons with recombination rates $\Gamma_D$ and $\Gamma_B$, respectively. $|D\rangle$ and $|B\rangle$ are thermally mixed with a two-LO-phonon process with rates $\gamma_\uparrow$ and $\gamma_\downarrow$. **c** Temperature dependence of the fast decay rate $\Gamma_S$ and **d** of the slow decay rate $\Gamma_L$. The plain curves are simulations with the two-phonon thermal mixing model, taking $\Gamma_B = 0.53$ ns$^{-1}$, $\Gamma_D = 0.015$ ns$^{-1}$, $\gamma_0 = 0.16$ ns$^{-1}$, $E_{LO1A} = 3.3$ meV, $E_{LO1B} = 3.6$ meV. **e** Temperature dependence of the fractions of emitted photons with fast and slow decay rates. The solid lines are derived from the three-level model with the same set of parameters. The error bars displayed in **c**, **d** and **e** represent the fitting errors of the parameters deduced from **a**

and two excitonic levels $|D\rangle$ and $|B\rangle$ representing, respectively, the dark and the bright excitons, with recombination rates $\Gamma_D$ (essentially nonradiative) and $\Gamma_B$ (mainly radiative). Thermal mixing of the excitonic states in NCs is commonly attributed to the emission and absorption of acoustic phonons from a mode whose energy matches the energy splitting between bright and dark excitons[79]. However, as shown in Supplementary Fig. 10, the temperature dependence of single FAPbI$_3$ NC PL decays cannot be reproduced by a one-phonon thermal mixing model. In fact, the temperature evolution of the long decay component (see Fig. 4d) is typical of an activation process with a characteristic temperature ∼ 40 K, corresponding to a thermal energy of ∼ 3.5 meV. This value is higher than the energies of acoustic phonons in these materials[39]. A further argument to rule out a contribution of acoustic phonons to the thermal mixing between fine structure states separated by few hundreds of µeV comes from the occurrence of a so-called "acoustic phonon bottleneck": as an acoustic wavelength cannot exceed the NC size, acoustic phonons with energies lower than 0.5 meV cannot develop in NCs with a size of ∼10 nm, given the average sound velocity $v_s$∼ 1270 m s$^{-1}$ in FAPbI$_3$[38,74].

As the activation energy matches the LO$_1$ phonon mode observed in the PL spectrum, we propose a two-phonon thermal mixing process involving the absorption and emission of LO$_1$ phonons (named LO$_{1A}$ and LO$_{1B}$) whose energy difference matches the bright−dark splitting (see Fig. 4b). The transition rates between the sublevels are $\gamma_\uparrow = \gamma_0 N_{LO1B}(N_{LO1A} + 1)$ and $\gamma_\downarrow = \gamma_0 N_{LO1A}(N_{LO1B} + 1)$, where $N_{LO1i} = 1/[\exp(E_{LO1i}/k_BT) - 1]$ are the Bose–Einstein phonon numbers (i = A, B), and $\gamma_0$ is a characteristic two-phonon mixing rate, which is discussed in the Supplementary Note 1 on the basis of Ref.[80] The decay rates $\Gamma_S$ and $\Gamma_L$ of the short and the long time components are deduced from the solutions of rate equations for the excitonic states populations:

$$\Gamma_{S,L} = \frac{1}{2}\left(\Gamma_B + \Gamma_D + \gamma_\uparrow + \gamma_\downarrow \pm \sqrt{\left(\Gamma_B - \Gamma_D + \gamma_\downarrow - \gamma_\uparrow\right)^2 + 4\gamma_\uparrow\gamma_\downarrow}\right)$$

(2)

Figures 4c, d show the evolution with temperature of $\Gamma_S$ and $\Gamma_L$ for the NC displayed in Fig. 4a. To reproduce the experimental temperature dependence of the short and the long decay components, as well as their relative weights, we chose two-optical phonon energies within the LO$_1$ sideband (centered around 3.5 meV). Their difference is explored between 200 and 500 µeV to match the dark-bright energy splitting. The determination of $\Gamma_D$ and $\Gamma_B$ is robust as these parameters are highly constrained by the values of the long and the short decay rates in the limits of low and high temperature regimes. Indeed, for $k_BT \ll E_{LO1i}$ (i = A,B), we have $\Gamma_S \rightarrow \Gamma_B$ and $\Gamma_L \rightarrow \Gamma_D$, whereas for $k_BT \gg E_{LO1i}$ (i = A,B), we have $\Gamma_S \rightarrow 2\gamma_0(k_BT)^2/(E_{LO1A}E_{LO1B})$ and $\Gamma_L \rightarrow (\Gamma_B + \Gamma_D)/2$. The parameter $\gamma_0$ is accurately adjusted to reproduce the evolutions with temperature of both $\Gamma_S$ and $\Gamma_L$. In Figs. 4c, d, these evolutions are well reproduced by Eq. 2, using the same set of parameters $\Gamma_B = 0.53$ ns$^{-1}$, $\Gamma_D = 0.015$ ns$^{-1}$, $\gamma_0 = 0.16$ ns$^{-1}$, $E_{LO1A} = 3.3$ meV, $E_{LO1B} = 3.6$ meV. Furthermore, Fig. 4e shows that the relative weights in photon numbers of the short and the long decay components are also in accordance with these parameters, assuming almost equal initial populations of bright and dark states after the laser pulse excitation and purely radiative and nonradiative recombinations, respectively. Measurements and analysis performed on five different FAPbI$_3$ NCs support our two-phonon thermal mixing model and lead to relaxation rates similar to those extracted from Fig. 4. It is worth noting that when we reverse the bright-dark level ordering in the model, the calculated temperature dependences of $\Gamma_S(T)$ and $\Gamma_L(T)$ remain very similar and reproduce as well the experimental data (see Supplementary Fig. 11). This is due to the fact that the energy of the LO phonons involved in the thermalization of these levels is much larger than their splitting. Hence, no conclusion can be derived on the strength of the Rashba effect in these materials[44,52–54,79], although a Rashba effect can be inferred from the non-centrosymmetry of the reported low temperature phase of FAPbI$_3$[51,68].

## Discussion

These spectroscopic investigations on single FAPbI$_3$ NCs demonstrate their suitability as efficient room temperature near-infrared

single-photon sources for quantum communication applications. The low temperature PL spectra reveal a sharp emission line, which undergoes spectral diffusion on a scale of a few meV, in stark contrast with the high spectral stability observed with CsPbBr$_3$ NCs[44,45,52–55]. Theoretical models of electric field fluctuations induced by the electric charge distributions of the FA cations should be developed to investigate their potential role in dephasing and spectral diffusion processes. The application of an external magnetic field leads to a broadening of the emission line and suggests that this line comprises an unresolved band-edge exciton fine structure on a sub-meV spectral range. The low temperature PL spectra of these NCs also unveil three LO optical phonon modes red-shifted by ~3.5 meV, ~11 meV and ~15 meV with respect to the ZPL. From the study of the temperature-dependent PL spectra of single NCs, we deduce that the contribution of the exciton–acoustic phonon to the thermal line broadening is negligible as weaker than 5 μeV K$^{-1}$. We find that the main contribution is due to Fröhlich coupling to a polar optical phonon mode identified in the PL spectra. This result points to the weakness of the carrier–phonon interactions related to the deformation potential due to acoustic phonons in comparison with the Fröhlich coupling with polar optical phonons. These conclusions are further supported by our investigations of the exciton recombination dynamics and relaxation rates within the band-edge exciton fine structure in these NCs. From the temperature dependence of the PL decay, we indeed find that the thermal mixing activation energy is higher than acoustic phonon energies and coincides with the lower-energy optical phonon mode whose energy is ~3.5 meV. The evolution of the PL decay with temperature can be reproduced with a two-LO–phonon thermal mixing model based on the absorption and the emission of optical phonons whose energy difference matches the expected sub-meV bright-dark splitting. The reduced thermal mixing at temperatures below 40 K may be interesting in the view of coherent spin manipulations. A more refined thermal mixing model should be developed to take into account all contributions of the various phonon modes and higher-order multi-phonon processes. Further studies will aim at identifying and reducing the low temperature dephasing and spectral diffusion processes in order to resolve the entire spectral fingerprint of the band-edge exciton fine structure of single NCs and reveal the bright-dark sublevel ordering. Overall, these spectroscopic findings will help to understand the carrier–phonon interaction mechanisms in FAPbI$_3$, such as the hot phonon bottleneck and the phonon glass character that affects the charge carrier transport in these perovskites. They are thus of prime importance to guide the development of next-generation devices for photovoltaics and for quantum technologies.

## Methods

**Preparation of FA oleate stock solution**. FA acetate (3.765 mmol, 0.392 g, Aldrich, 99%) was loaded into a 50 mL three-neck flask along with octadecene (ODE, 18 mL) and oleic acid (12 mL, Sigma-Aldrich, 90%). The reaction mixture was degassed three times at room temperature, heated to 100 °C under nitrogen until the reaction was completed and cooled down to room temperature. The obtained solution was stored in a glovebox.

**Synthesis of FAPbI3 NCs**. In a 25 mL three-necked flask, lead (II) iodide (61.1 mg, 0.133 mmol, Sigma-Aldrich) was suspended in ODE (4.6 mL), heated to 60 °C and then dried under vacuum for 30 min. Subsequently, the reaction mixture was heated to 110 °C under nitrogen, followed by the syringe addition of dried solvents—oleylamine (0.5 mL, STREM) and oleic acid (1.0 mL). Once the lead (II) iodide was dissolved, the reaction mixture was cooled to 80 °C. At this point, the mixture of FA oleate stock solution (5.0 mL) and ODE (1.0 mL) was injected into the reaction flask. After another 15 s, the reaction mixture was cooled by a water-ice bath.

**Isolation and purification of NCs**. The crude solution was centrifuged at 12.1 krpm for 5 min (Centrifuge: Eppendorf 5804) and the supernatant was discarded. The precipitate was dissolved in hexane (0.3 mL) and the solution was centrifuged again (12.1 krpm, 5 min). The supernatant, containing monodisperse NCs, was retained for the subsequent purification, whereas the precipitated NCs were discarded. In order to remove the excess of the organic ligands, hexane (0.9 mL), toluene (0.6 mL) and methyl acetate (1.95 mL) were added to the colloidal solution of NCs and the resulting solution was centrifuged at 13.4 rpm for 3 min. The supernatant was discarded and the precipitate was dissolved in toluene (1 mL), yielding concentration of ca. 10–15 mg/mL. This solution was additionally filtered through a 0.45 μm PTFE-filter prior to optical measurements.

**Sample characterization**. Ensemble ultraviolet-visible absorption spectra were collected using a Jasco V770 spectrometer operated in transmission mode. Fluoromax iHR 320 Horiba Jobin Yvon spectrofluorimeter equipped with a photomultiplier was used to acquire steady-state PL spectra from solution. The excitation wavelength was 400 nm, provided by a 450 W Xenon lamp dispersed with a monochromator. Measured intensities were corrected to take into account the spectral response of the detector. Transmission electron microscopy images were collected using a JEOL JEM-2200FS microscope operated at 200 kV. Powder X-ray diffraction patterns were collected with a STOE STADI P powder diffractometer, operating in transmission mode. A germanium monochromator, Cu Kα1 irradiation and a silicon strip detector (Dectris Mythen), were used.

**Single NC spectroscopy**. The NCs in toluene were kept under N$_2$ atmosphere for further analysis. For single NC measurements, a dilute solution of FAPbI$_3$ NCs was mixed with a 1 wt% poly(methyl methacrylate) solution and spin-coated onto clean glass or sapphire coverslips with a rotation speed of 2000 rpm. Single NC measurements at room temperature were conducted using a customized confocal scanning fluorescence microscope. It is based on a commercial microscope (Nikon Eclipse Ti) and a 1.45 numerical aperture oil-immersion objective. The emitted photon stream is filtered by a long-pass filter (593 nm blocking edge), then split by a 50/50 beam splitter and sent to two avalanche photodiodes. For wide-field imaging, an additional lens is used to focus the excitation beam on the back focal plane of the objective and the emitted photons are sent to a CCD camera (Princeton Instruments, ProEM 512). For the low temperature measurements, a home-built scanning confocal microscope based on a 0.95 numerical aperture objective is placed in a cryostat and used to image single NCs by raster scanning the sample. The NCs are excited at 488 nm or 727 nm with a laser passed through a clean-up filter. The emitted photons are filtered by a long-pass filter (776 nm blocking edge) and sent to a single-photon counting avalanche photodiode and a spectrograph. PL decays are recorded with a conventional time correlated single-photon-counting setup using a pulsed laser source (optical parametric oscillator at 488 nm, 200 fs pulse width, with a repetition rate reduced to 8 MHz with a pulse-picker).

**Data availability**. All relevant data that support our experimental findings are available from the corresponding author upon reasonable request.

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

## Acknowledgements

We acknowledge the financial support from the French National Agency for Research, Région Aquitaine, Idex Bordeaux (LAPHIA Program), the French Ministry of Education and Research and the Institut universitaire de France. M.I.B. acknowledges financial support from the Swiss National Science Foundation (SNF Ambizione grant, grant no. PZENP2_154287). M.V.K. acknowledges financial support from the European Research Council under the European Union's Seventh Framework Program (ERC Starting Grant NANOSOLID, Grant Agreement No. 306733). M.I.B. and M.V.K. are grateful to Empa Electron Microscopy Center for access to the instruments and for technical assistance.

## Author contributions

M.I. B. and M.V.K. prepared the samples and performed their ensemble characterization. M.F. and P.T. performed the optical experiments. B.L., P.T. and J.E. interpreted the data, which were analyzed by M.F., J.-B.T, P.T. and B.L. B.L. and P.T. developed the two-phonon-process thermalization model. P.T., J.E. and B.L. wrote the manuscript with input from all authors. B.L. supervised the project.

## Additional information

**Competing interests:** The authors declare no competing interests.

