## [Peer Review File · Nature Communications]

Reviewers' comments:

Reviewer #1 (Remarks to the Author):

In this manuscript, the authors report single-dot studies of organic-inorganic lead halide FAPbI₃ nanocrystals (NCs) that are designed to emit near-infrared luminescence. First, they observed near-complete photon-antibunching from a single FAPbI₃ NC, which confirms them as infrared single-photon sources at room temperature, emphasizing the possible future application in quantum network science. Then, single-particle spectroscopy was performed at cryogenic temperature to study the electronic structures and, in particular, exciton-phonon interaction of these NCs. Several phonon modes were well resolved from single-dot PL spectra and their origin were reasonably well explained based on a broad search on the literatures. The observation about negligible (or no) acoustic phonon modes in these NCs is quite different from what we normally know for most of NCs, which I think invoked them to develop two-phonon thermal mixing model to explain temperature-dependent PL dynamics. It is also significant to reveal the exotic temperature dependence of PL energy shift for the same single NC over a large range of environmental temperature. Exciton fine structure was examined from PL decay dynamics of a single FAPbI₃ NC as the temperature is varied, which clearly shows two different decay pathways (bright and dark excitons) from the lowest energy levels in the exciton fine structure. They provided a model that uses thermal mixing to LO-phonon instead of acoustic phonon coupling that agrees well with experimental data.

In my opinion, this work is well done both in the experiment and the analysis, providing a good quality of experimental data and a supporting model. However, my concern is that this work is specific just to one kind among many families of perovskite NCs. I am not sure their unique observations on individual FAPbI₃ NCs, such as no presence of acoustic phonon modes and exciton fine structure with a very small bright-dark exciton separation, can be applied (or generalized) to other kinds with different components for organic-part and/or halide-part, for examples. It would be great if this work can be expanded including, at least, different halide components (FAPbX₃, X=I, Br). So, I do not recommend the current manuscript to be published in Nature Communication. Below are a few comments and questions.

1. Lorentzian or Gaussian fit to the single NC spectrum: these two functions are used to fit the single NC spectrum both at room and cryogenic temperature, but there is no explanation about which curve fit should be used. It has to be one function through the manuscript with a proper reason.
2. In Fig. 1e and 1f: a double-Lorentzian curve is used to fit the spectral profile, and one Lorentzian curve with a smaller amplitude (not shown in Fig. 1e) is assigned to the phonon band. But, based on the blinking trace in Fig. 1f, the NC switches ON-OFF very often, implying the NC is charged for a fairly long time. Therefore, double-Lorentzian feature could be from neutral and charged NC PL with a possible combination of phonon mode.
3. It would be nice to provide the statistics of the $g(2)(0)$ values.
4. In page 6, it says "we have investigated the PL spectrum of more than a hundred of single NCs at cryogenic temperatures". But, I see only ~ 25 NCs in the histogram (Fig. 2e). Please, include all data to improve the statistics, if allowed. This will provide a more accurate estimation on the value of LO-phonon energies.
5. Single NC linewidth is as narrow as 1.2 meV at 3.6K, which is not radiative-limited. Since there is no linewidth broadening due to acoustic phonon mode and bright-dark exciton splitting is very small (0.3 meV in Fig. 4), how can this linewidth be explained? What will be the limit of line-broadening in case of no acoustic phonons?
6. In page 12, please describe how the thermal activation energy of ~ 4 meV was inferred.

Reviewer #2 (Remarks to the Author):

Report on the paper entitled “Unravelling exciton-phonon coupling in individual FAPbI₃ nanocrystals emitting near-infrared single photons” by M. Fu *et al.*

Thin films of hybrid organic-inorganic lead halide perovskite have attracted in recent years a huge interest due to their exceptional optical properties that make possible the development of highly efficient photovoltaic and optical devices based on this materials. Inspired by this success, in 2016 and 2017 the Kovalenko's group have synthesized stable organic-inorganic nanocrystals (NCs). In this manuscript, the authors report on spectroscopic studies of single FAPbI₃ NCs avoiding in this way the ensemble average effect. The authors obtained three main results: a) a strong photon antibunching at room temperature $g^2(0)=0.1$, b) the experimental identification of LO phonons in the photoluminescence (PL) spectra of a single NC at low temperature and the determination of the Frölich mechanisms as the main exciton-phonon interaction responsible for emission line broadening c) the experimental evidence that the PL dynamics is governed by a thermal mixing between dark and bright exciton fine structure states. Authors propose a two-optical phonon Raman scattering process at the origin of the thermal mixing. The whole ensemble of these results is of great relevance. A detailed knowledge of the band-edge exciton states, their dynamics and their interaction with phonons is of prime importance to understand the exceptional optical properties of perovskite materials.

Hence I would recommend its publication after the authors address the following questions and minor remarks.

Questions:

- 1) The 3K PL line is surprisingly broad for a single NC ($\approx 1\text{meV}$), authors invoke the charge fluctuations induced by residual cation movements and the fact that this line probably hide an unresolved band edge exciton fine structure as observed in other inorganic lead halide perovskite NCs to explain this broadening.
 - a. It is possible to quantify or estimate the expected broadening due to charge fluctuations?
 - b. Authors give a very general scheme of the exciton fine structure consisting into two levels, one bright and the other dark. They claim to a sub-meV splitting between them. Could the authors give more quantitative information on the expected fine structure of FAPbI₃ NCs (one, two or three bright states) and estimate the splitting energies between different exciton states on the base of the given references: 50-54?. There is also another reference that should be added to this discussion: Ramade J. et al. *Nanoscale* 10, 6393 (2018).
 - c. In references 50-54 the observed excitonic lines show doublets or triplets with crossed linear polarizations. Have the authors tried to isolate different contributions to the exciton fine structure by means of a polarization analysis of the PL?
- 2) Recently a great attention has been focused on the exciton fine structure of the halide perovskites (ref 50-54). In particular, authors in reference 53 propose that Rashba effect might play a main role to determine exciton fine structure: Rashba effect combined to strong spin-orbit coupling must lead to a bright triplet exciton state placed at lowest energy. Chen L. and co-workers (*Nano Letters* 18, 2071, (2018)) have

recently shown that the PL dynamics of an ensemble of perovskite NC is governed by thermal mixing of exciton fine structure states and, from these experimental data, these authors unequivocally place the dark state at lowest energy regardless of the type of material, i.e., inorganic or organic-inorganic perovskite NCs. Ramade J. and co-workers (Nanoscale 10, 6393 (2018)) from spectroscopic data of single CsPbBr₃ NCs and theoretical estimations conclude that the lowest exciton state in this material is dark. In order to obtain a dark exciton state at higher energy than the bright exciton states a lack of inversion symmetry or centrosymmetry breaking on a local scale is needed. In organic-inorganic halide perovskites dynamic fluctuations of the local structure can help to trigger the Rashba effect. Authors of this manuscript show in Figure 4 b) a dark exciton state at lowest energy. Can authors show which of two possibilities, described above, better explains their experimental results? Could the authors comment on that?

- 3) In page 12 authors explain that they have applied an external magnetic field up to 7 T « to unveil the dark state emission line » but they did not observe any new PL line. They observed only a line broadening and they concluded that dark exciton line is very closed to bright exciton (their splitting is in sub-meV range). However, authors do not comment about the changes observed in the PL decay under magnetic field that has to be also affected by « brightening effect » of the dark exciton.
- 4) Could the authors give a more quantitative description of the two-phonon thermal mixing process between bright and dark exciton states?. For example, is it possible to link γ_0 rate to the electron phonon coupling strength, phonon dispersion.... ? γ_0 is a spin flip rate. Can authors explain why this two-phonon process is more probable for LO₁ than for LO₂ and why this process can mix a dark and a bright state?. Can authors comment on the polarization properties of the dark state PL, will be it modified by this process?.
- 5) Parameters obtained from fitting of PL decay curves for different temperatures should be comment in more detail.

Minor remarks:

- 1) X-ray diffraction and spectroscopic data of Figure 1 are obtained at room temperature. I suggest to the authors to clearly give this information in the figure caption or in the different panels of this figure. The same for low temperature data.
- 2) At the end of page 4, the bulk bandgap of FAPbI₃ is given to be equal to 1.47 eV, i.e., 230 meV lower in energy than the emission of NCs. Authors explain that as a weak quantum size-effect in the studied NCs that have a size 2—3 times larger than the exciton Bohr radius. Another possible explanation is that measuring of the PL spectra very often does experimental determination of bandgap. In bulk materials PL spectra is some times dominated by bound excitons and not free excitons and bound excitons appear at lower energy than free excitons.
- 3) In Figure 3 a) authors give in blue the fitted spectra by using a triple Gaussian or Lorentzian profiles. From 55K until 150K it will be interesting to have in this

figure the three Lorentzian or Gaussian curves used to fit as well as the whole fitted spectra. The same for similar figures in the supplementary information.

- 4) In figure 4 a) it will be suitable to give the experimental data in a log-log representation in which the bi-exponential behaviour will be more evident. It will be interesting also to have the fitting curve superimposed on the experimental data.
- 5) In figure 4 d) could authors give the errors bars for Γ_L ?
- 6) Authors should also add two references directly related with topics addressed in this work: fine structure- Ramade J. et al. *Nanoscale* 10, 6393 (2018), and dynamics of exciton PL in ensembles of perovskite NCs ; Chen L. *Nano Letters*. 18, 2071, (2018).

In the following, we answer all the comments of the reviewers.

Reviewer #1

We thank the reviewer for his positive comments on the quality of our experiments and their analysis. We first address the reviewer's main concern.

“However, my concern is that this work is specific just to one kind among many families of perovskite NCs. I am not sure their unique observations on individual FAPbI₃ NCs, such as no presence of acoustic phonon modes and exciton fine structure with a very small bright-dark exciton separation, can be applied (or generalized) to other kinds with different components for organic-part and/or halide-part, for examples. It would be great if this work can be expanded including, at least, different halide components (FAPbX₃, X=I, Br). So, I do not recommend the current manuscript to be published in Nature Communication.”

Studying lead halide perovskite NCs with single NC spectroscopy has recently emerged as a unique tool to reveal optical properties that were hidden in bulk or ensemble studies. Removing extrinsic effects and ensemble averaging offers a wealth of information on the quantum nature of the emitted light, the band-edge exciton fine structure and the relaxation dynamics within the sublevels, as well as the detailed exciton-phonon coupling mechanisms and strengths. However, these experimental studies are demanding since many single NCs need to be carefully studied before a global and robust interpretation of the data can be carried out. Our conclusions are based on months of experimental work, and thus a whole study of various perovskite NCs with different compositions is beyond the scope of this work. We would like to mention that similar spectroscopic studies on FAPbBr₃ NCs, synthesized in the group of M. Kovalenko, are led in parallel with this work in another group of experimentalists and is now on the verge of being submitted.

Recently, Chen et al. (Nano Letters 18, 2071, (2018)) published a time-resolved magneto-photoluminescence spectroscopic study of *ensembles* of inorganic lead halide perovskite NCs with different compositions at cryogenic temperatures. The overall behavior of the temperature-dependent PL is similar to our observations. Importantly, as commonly done, these authors base their interpretation on the one-acoustic-phonon thermal mixing model that we introduced 15 years ago for usual II-VI and III-V semiconductor NCs (Labeau et al., Phys. Rev. Lett. 2003). This yields phonon energies that are much larger than the expected - and sometimes observed - fine structure splittings. Here we show that for the soft perovskite materials the acoustic phonons do not interact with excitons. They are not well defined due to the strong lattice anharmonicity. The concept of phonon glassy state is discussed in the manuscript. Moreover, we show that the common one-phonon thermal mixing model is not valid in these materials (see below). We think that our two-optical-phonon model is of great relevance to interpret the temperature-dependent properties of these perovskite NCs, and will be emulated by the community.

With these elements, we hope to convince the reviewer that our manuscript is far-reaching and provides important advances in the physics of lead halide perovskite NCs that are worthy of publication in Nature Communication.

We address below the questions and comments raised by the reviewer.

1- *“Lorentzian or Gaussian fit to the single NC spectrum: these two functions are used to fit the single NC spectrum both at room and cryogenic temperature, but there is no explanation about which curve fit should be used. It has to be one function through the manuscript with a proper reason.”*

Response:

We thank the reviewer for giving us the opportunity to clarify this point. At low temperature (from 3.5 K to 30 K), the ZPL profile is found to have a Gaussian shape, while at higher temperature (up to room temperature) the line is better reproduced with a Lorentzian shape. In fact the emission line profile results from the convolution of two broadening mechanisms: spectral diffusion and thermal dephasing. Below 30 K thermal dephasing is negligible and the lineshape is essentially set by the distribution of emission line frequencies sampled during the integration time of the PL spectrum. The random processes at the origin of spectral diffusion lead to a Gaussian-shaped emission line profile. Above 30 K, the homogeneous line broadening by thermal dephasing prevails over the spectral excursions caused by spectral diffusion. The homogeneous line thus retrieves its genuine Lorentzian shape.

In the revised manuscript, we explain the use of Gaussian fits when we discuss the low temperature spectra (page 7): *“(…) The line shape is thus essentially set by the distribution of emission line frequencies sampled during the integration time of the spectra. The random process at the origin of spectral diffusion leads to a Gaussian-shaped emission line.”*

In the paragraph dedicated to the discussion of the thermal broadening of the emission line (page 11), we added one sentence: *“Above 30 K, the homogeneous line broadening by thermal dephasing prevails over the spectral excursions caused by spectral diffusion. The emission line thus acquires a Lorentzian shape that is characteristic of a homogeneous line at higher temperatures, as exemplified in Fig. 3a.”*

We also added in the caption of Fig. 3: *“The spectra are fitted (blue lines) with Gaussian profiles up to 30 K and with Lorentzian profiles above 30 K (see text)”*.

2- *“In Fig. 1e and 1f: a double-Lorentzian curve is used to fit the spectral profile, and one Lorentzian curve with a smaller amplitude (not shown in Fig. 1e) is assigned to the phonon band. But, based on the blinking trace in Fig. 1f, the NC switches ON-OFF very often, implying the NC is charged for a fairly long time. Therefore, double-Lorentzian feature could be from neutral and charged NC PL with a possible combination of phonon mode.”*

Response:

We agree with the reviewer on this point. However, since we did not observe any spectral signature of the trion in the low-temperature spectra of single NCs, we did not mention its possible contribution to the lineshape of the room temperature spectra.

Moreover, from our study of the PL spectra of single NCs as a function of temperature, the red-shoulder observed in the PL spectrum of single NCs at room temperature is consistent with an optical phonon sideband whose energy lies in the range 10-15 meV, as suggested by the histogram of Fig. 2e. Nevertheless, since we cannot completely rule out a residual contribution from trions emission to the spectra, we added in the text (page 6): *“The asymmetry displayed in the PL spectral profile is the hallmark of an optical phonon sideband, although a residual contribution from a low emissive state to the red shoulder cannot be ruled out.”*

3- *“It would be nice to provide the statistics of the $g^{(2)}(0)$ values.”*

Response:

We indeed measured the PL intensity autocorrelation function for 22 NCs. We find values of $g^{(2)}(0)$ distributed between 0.05 and 0.3. Examples of autocorrelation histograms are displayed in the appended figure, together with the histogram of the $g^{(2)}(0)$ values. This figure has been added to the Supplementary file (Fig. S1) and we refer to this figure in the revised manuscript.

Fig. S1: Fluorescence intensity autocorrelation function of several FAPbI₃ single NCs at room temperature. (a-k) Histograms of time delays between consecutive photon pairs from the PL of individual FAPbI₃ NCs. (l) Distribution of $g^{(2)}(0)$ values. The main contribution to $g^{(2)}(0)$ is attributed to residual biexciton radiative recombination.

4- “In page 6, it says “we have investigated the PL spectrum of more than a hundred of single NCs at cryogenic temperatures”. But, I see only ~ 25 NCs in the histogram (Fig. 2e). Please, include all data to improve the statistics, if allowed. This will provide a more accurate estimation on the value of LO-phonon energies.”

Response:

We have indeed recorded the PL spectra of more than a hundred single NCs at cryogenic temperature. However, the histogram of Fig. 2e was built out of 24 NCs. Indeed we selected the most stable NCs, which do not show spectral jumps during the integration time, in order to avoid broadening of the spectra and to ensure an accurate determination of the LO-phonon energies. We now improved the statistics of LO phonon energies (32 NCs). Moreover, we took this opportunity to add the histogram of

LO₃ phonon energies in Fig. 2e of the revised manuscript (see the appended histogram).

5- “Single NC linewidth is as narrow as 1.2 meV at 3.6K, which is not radiative-limited. Since there is no linewidth broadening due to acoustic phonon mode and bright-dark exciton splitting is very small (0.3 meV in Fig. 4), how can this linewidth be explained? What will be the limit of line-broadening in case of no acoustic phonons?”

Response:

A linewidth of 1.2 meV is indeed not limited by an exciton recombination lifetime in the ns range. It is also larger than the resolution of our spectrograph, which can be as low as 120 μ eV (with the 1800 lines/mm grating, see Fig. S3). No thermal broadening of the ZPL is observed in the 3-30 K temperature range, which indicates that the contribution of phonons to the line broadening is negligible. Below 30 K, the line broadening is thus due to spectral diffusion that occurs during the integration time of the PL spectrum. This is consistent with the observation of larger linewidths when increasing the integration time of the PL spectrum.

In the revised manuscript we rephrased the text to improve the clarity of the explanation (page 7): “*The ZPL undergoes a small spectral diffusion with an excursion contained within a few meV (see the Supplementary Fig. S2). This behavior stands apart from that of fully inorganic perovskite NCs, which display a remarkable spectral stability in the low temperature PL spectra. A possible origin of the spectral diffusion can be explained as follows. Contrary to cations in fully inorganic perovskites, organic cations in hybrid lead-halide perovskites are known to have electric dipole and quadrupole moments. Photo-generated charge carriers cause the neighboring cations to reorient and locally align their dipoles to form ferroelectric domains and maximize the screening effect. This results in a change of the electrostatic potential in the NC, leading to a shift of the transition energy of the exciton. It is consistent with the recently reported locally disordered low temperature phase of FAPbI₃. During the acquisition*

of the PL spectra, random configurations of cation dipole organizations (domain size and orientation) will be sampled and lead to spectral diffusion of the exciton recombination line. The line shape is thus essentially set by the distribution of emission line frequencies sampled during the integration time of the spectra. The random process at the origin of spectral diffusion leads to a Gaussian-shaped emission line. The sharpest lines are obtained when reducing the integration time as well as the excess energy between the excitation photons and the emission photons. Linewidths as narrow as 0.8 meV can be obtained when recording the spectra over a very short integration time (0.2 s) and by using an excitation wavelength in the near infrared (~ 730 nm) (see Fig. 2 and the Supplementary Fig. S2).”

6- *“In page 12, please describe how the thermal activation energy of ~ 4meV was inferred”*

Response:

We thank the reviewer for giving us the opportunity to clarify this point. The temperature evolution of the long decay component displayed in Fig. 4d is characteristic of an activation process with a characteristic temperature of ~40 K, corresponding to a thermal activation energy of ~3.5 meV in the Bose-Einstein function. In the revised manuscript, we modified this sentence (page 14): *“In fact, the temperature evolution of the long decay component (see Fig. 4d) is typical of an activation process with a characteristic temperature ~ 40 K, corresponding to a thermal energy of ~3.5 meV. This value is higher than the energies of acoustic phonons in these materials”*.

Reviewer #2

We thank the reviewer for extremely positive comments on the great relevance of the whole ensemble of our results, and for recommendation for publication after addressing the following questions and minor remarks.

1- *“The 3K PL line is surprisingly broad for a single NC ($\approx 1\text{meV}$), authors invoke the charge fluctuations induced by residual cation movements and the fact that this line probably hide an unresolved band edge exciton fine structure as observed in other inorganic lead halide perovskite NCs to explained this broadening.*

The reviewer raises an important point. Indeed, studies of the composition-related emission linewidth broadening by comparing fully inorganic and hybrid organic-inorganic perovskite NCs are crucial to reach better understanding of the fundamental properties of these materials and for the development of applications (e.g. improvement of their color purity in light emitting devices).

a. It is possible to quantify or estimate the expected broadening due to charge fluctuations?

Response:

In usual NCs such as CdSe NCs, an estimation of the spectral fluctuations can be obtained using known Stark coefficients of the NCs and assuming a charge displacement at the surface of the NC (see for example Fernée et al., Nanotechnology **24** (2013) 465703). Unfortunately, similar estimations cannot be carried out in perovskites because the Stark properties are not known and the spectral fluctuations are of different nature. Unlike CdSe, perovskite are soft materials where structural rearrangements can occur. Moreover, organic cations in hybrid lead-halide perovskites are known to have electric dipole and quadrupole moments (Chen et al., PNAS 2017). A likely explanation of the spectral diffusion is based on the following scenario. The photo-generated charge carriers cause the proximal cations to reorient (Taylor et al. JPCL 2018) and locally align their dipoles to form ferroelectric domains (Frost et al. NanoLett. 2014) and maximize the screening effect. This results in a change of the electrostatic environment in the NC, leading to a shift of the transition energy of the exciton. During the acquisition of the PL spectra, random configurations of cation dipoles reorganizations (domain size and orientation) are sampled and lead to spectral diffusion of the exciton recombination line.

The quantitative estimation of the fluctuations need specific measurements (e.g. Stark measurements, dynamics of the cation dipoles domains,...) and advanced theoretical modeling. Such a work is far beyond the scope of this paper. In the revised manuscript, we added this discussion of the origin of the spectral diffusion (page 7):

“The ZPL undergoes a small spectral diffusion with an excursion contained within a few meV (see the Supplementary Fig. S2). This behavior stands apart from that of

fully inorganic perovskite NCs, which display a remarkable spectral stability in the low temperature PL spectra. A possible origin of the spectral diffusion can be explained as follows. Contrary to cations in fully inorganic perovskites, organic cations in hybrid lead-halide perovskites are known to have electric dipole and quadrupole moments. Photo-generated charge carriers cause the neighboring cations to reorient and locally align their dipoles to form ferroelectric domains and maximize the screening effect. This results in a change of the electrostatic potential in the NC, leading to a shift of the transition energy of the exciton. It is consistent with the recently reported locally disordered low temperature phase of FAPbI₃. During the acquisition of the PL spectra, random configurations of cation dipole organizations (domain size and orientation) will be sampled and lead to spectral diffusion of the exciton recombination line. The line shape is thus essentially set by the distribution of emission line frequencies sampled during the integration time of the spectra. The random process at the origin of spectral diffusion leads to a Gaussian-shaped emission line.”

b. Authors give a very general scheme of the exciton fine structure consisting into two levels, one bright and the other dark. They claim to a sub-meV splitting between them. Could the authors give more quantitative information on the expected fine structure of FAPbI₃ NCs (one, two or three bright states) and estimate the splitting energies between different exciton states on the base of the given references: 50-54?. There is also another reference that should be added to this discussion: Ramade J. et al. Nanoscale 10, 6393 (2018).

Response:

We thank the reviewer to attract our attention on the interesting work of Ramade et al. (Nanoscale 2018) on single CsPbBr₃ NCs, that we now cite in the revised manuscript, together with the references related to the spectroscopic studies of the band-edge exciton fine structure in single cesium lead halide NCs (page 7, end of the first paragraph).

In particular, for CsPbI₃ NCs, statistics of fine-structure splitting measured from the low-temperature PL spectra of ~80 single NCs indicate an average splitting value of $356 \pm 108 \mu\text{eV}$ (see Fig. 1d in Yin et al., Phys. Rev. Lett. 119 (2017) 026401). For FAPbI₃ NCs, there is no experimental report on the band-edge exciton fine structure. As for other lead halide NCs, the fine structure is expected to comprise a dark singlet and a bright triplet, whose degeneracy may be lifted by crystal field, depending on the crystal structure. The fine structure splitting should be similar to that of CsPbI₃ NCs, since it should not be significantly influenced by the nature of the cation. Furthermore, Becker et al. (Nature 2018, Supplementary Information) calculated a singlet-triplet splitting of ~100 μeV for CsPbI₃ NCs, based on the exchange interaction.

The temperature dependence of the PL decay can be reproduced with a simple model assuming a thermal mixing between two fine structure energy levels: one related to the dark state and the other to the bright triplet manifold. The assumption of a sub-

meV bright-dark splitting is supported by two experimental observations:

i) The sharpest emission lines in the PL spectra of single NCs display a linewidth of ~ 0.8 meV which is not limited by the resolution of the spectrograph (our spectral resolution being of ~ 120 μeV). Therefore, unresolved spectral components associated to fine structures sublevels may lie within the linewidth.

ii) The emission linewidth is broadened under magnetic fields, which suggests magnetic splittings and brightening within the linewidth.

In order to justify the choice of a sub-meV bright-dark splitting in our thermal mixing model, we rephrased the first paragraph of the section describing the evolution of the PL decays, in page 13:

“... , the PL decay displays a clear biexponential character with a long decay component that gains weight and shortens with temperature, while the short component shortens and loses weight until it vanishes above 70 K. Moreover, under magnetic fields, the PL decay shows a shortening of its long-lived component with an increase of its weight (See the Supplementary Fig. S8). These characteristic behaviors are obvious signatures of thermal mixing and magnetic coupling between bright and dark fine structure sublevels, and are similar to that previously reported in CdSe NCs. For FAPbI₃ NCs, no spectroscopic study of the band-edge exciton fine structure is reported. Nevertheless, this fine structure should be weakly influenced by the nature of the cation, and thus be similar to that of CsPbI₃. It comprises a bright triplet, whose degeneracy may be lifted by crystal field depending on the crystal structure and a dark single state. Fine-structure splitting measurements, based on PL spectra of single CsPbI₃ NCs, led to a statistical distribution with an average splitting of ~ 300 μeV and an upper value of ~ 500 μeV . To unveil a dark state emission line via magnetic brightening, we recorded PL spectra of individual FAPbI₃ NCs under external magnetic fields up to 7 T. We did not observe any emerging peak but a slight emission line broadening (Supplementary Fig. S9). This suggests that the dark exciton level lies close to a bright level, with a bright-dark energy separation within a linewidth, i.e. in the range of few hundred μeV .

The temperature dependence of the PL decay can be reproduced with a simple model assuming a thermal mixing between two fine structure energy levels: one related to the dark state and the other to the bright triplet manifold.”

c. In references 50-54 the observed excitonic lines show doublets or triplets with crossed linear polarizations. Have the authors tried to isolate different contributions to the exciton fine structure by means of a polarization analysis of the PL?

Response:

We did not investigate the polarization properties of the PL spectra. Due to fast spectral diffusion of these NCs, the spectral components of the band-edge exciton fine structure are buried in the emission line.

2- *“Recently a great attention has been focused on the exciton fine structure of the*

halide perovskites (ref 50-54). In particular, authors in reference 53 propose that Rashba effect might play a main role to determine exciton fine structure: Rashba effect combined to strong spin-orbit coupling must lead to a bright triplet exciton state placed at lowest energy. Chen L. and co-workers (Nano Letters 18, 2071, (2018)) have recently shown that the PL dynamics of an ensemble of perovskite NC is governed by thermal mixing of exciton fine structure states and, from these experimental data, these authors unequivocally place the dark state at lowest energy regardless of the type of material, i.e., inorganic or organic-inorganic perovskite NCs. Ramade J. and co-workers (Nanoscale 10, 6393 (2018)) from spectroscopic data of single CsPbBr₃ NCs and theoretical estimations conclude that the lowest exciton state in this material is dark. In order to obtain a dark exciton state at higher energy than the bright exciton states a lack of inversion symmetry or centrosymmetry breaking on a local scale is needed. In organic-inorganic halide perovskites dynamic fluctuations of the local structure can help to trigger the Rashba effect. Authors of this manuscript show in Figure 4 b) a dark exciton state at lowest energy. Can authors show which of two possibilities, described above, better explains their experimental results? Could the authors comment on that?"

Response:

We thank the reviewer for giving us the opportunity to discuss this point in more details. First of all, we would like to stress that, as commonly done, Chen and co-workers (Nano Letters 18, 2071, (2018)) founded their interpretation on the model of thermal mixing with single acoustic phonons, taking the dark state below the bright one (Labeau et al., Phys. Rev. Lett. 2003). From the temperature dependence of the ensemble averaged $\Gamma_L(T)$, they obtained phonon energies E_{Ac} in the range 4-14 meV, depending on the perovskite chemical composition, and associated these energies to the bright-dark splitting. They also deduced that their data cannot be explained if the dark state is above the bright one. However, their conclusions are in contradiction with the expected exciton fine structure splittings (see response 1-b), DFT calculations of the acoustic phonon band structure (e.g. Yang et al. Nature Comm. **8** (2017) 14120) and the low exciton-acoustic phonon couplings extracted from the temperature dependence of ZPL linewidths measured on single NCs. Indeed, the acoustic phonon modes are not well defined in these soft materials due to the strong lattice anharmonicity, a signature of acoustic phonon glass character.

Single NCs data give access to the temperature dependence of both $\Gamma_S(T)$ and $\Gamma_L(T)$. This allows to constrain the relaxation rates used in the simulations. Indeed, for the single phonon model, in the low temperature regime where $k_B T \ll E_{Ac}$, we have

$$\Gamma_S \rightarrow \Gamma_B + \gamma_0^{Ac} \quad \text{and} \quad \Gamma_L \rightarrow \Gamma_D \quad (\text{when the dark level lies below the bright}).$$

In the high temperature regime where $k_B T \gg E_{Ac}$, we have

$$\Gamma_S \rightarrow 2\gamma_0^{Ac} (k_B T / E_{Ac}) \quad \text{and} \quad \Gamma_L \rightarrow (\Gamma_B + \Gamma_D) / 2 .$$

As clearly shown in the appended figure, single NC data cannot be reproduced with this model. Indeed, a set of parameters can be found to reproduce $\Gamma_S(T)$ or $\Gamma_L(T)$, but no set of parameters can be found to reproduce both.

Fig. S10. Attempts to reproduce the temperature dependence of the PL decay with the one-phonon thermal mixing model. In this model, emission and absorption of acoustic phonons from a mode whose energy E_{Ac} matches the bright-dark splitting induces a thermalization of the excitonic states. The transition rates between bright and dark levels are $\gamma_{\uparrow} = \gamma_0^{Ac} N_B$ and $\gamma_{\downarrow} = \gamma_0^{Ac} (N_B + 1)$, where $N_B = 1/[\exp(E_{Ac}/k_B T) - 1]$ is the Bose-Einstein phonon number at the temperature T . The decay rates of the short (blue points) and the long (red points) components are the same as in Fig. 4. In the simulations (solid curves), the dark level is placed below (a and b) or above (c and d) the bright level. The temperature dependence of the short and the long decay components cannot be reproduced with the same set of parameters, whatever the bright-dark level ordering. (a) The evolution of the short component is reproduced with the parameters $\Gamma_B = 0.3 \text{ ns}^{-1}$, $\Gamma_D = 0.01 \text{ ns}^{-1}$, $\gamma_0^{Ac} = 0.2 \text{ ns}^{-1}$, $E_{Ac} = 3 \text{ meV}$. (b) The evolution of long component is reproduced with the parameters $\Gamma_B = 0.8 \text{ ns}^{-1}$, $\Gamma_D = 0.01 \text{ ns}^{-1}$, $\gamma_0^{Ac} = 1 \text{ ns}^{-1}$, $E_{Ac} = 5 \text{ meV}$ (c) The evolution of the short component is reproduced with the parameters $\Gamma_B = 0.5 \text{ ns}^{-1}$, $\Gamma_D = 0.001 \text{ ns}^{-1}$, $\gamma_0^{Ac} = 0.35 \text{ ns}^{-1}$, $E_{Ac} = 5 \text{ meV}$. (d) No set of parameters can be found to reproduce the evolution of the long component. This evolution is very roughly approached with the parameters $\Gamma_B = 0.8 \text{ ns}^{-1}$, $\Gamma_D = 0.001 \text{ ns}^{-1}$, $\gamma_0^{Ac} = 0.02 \text{ ns}^{-1}$, $E_{Ac} = 0.5 \text{ meV}$.

This figure has been added to the Supplementary file (Fig. S10) and we refer to it in the revised manuscript (page 14): “However, as shown in the Supplementary Fig. S10, the temperature dependence of single FAPbI₃ NC PL decays cannot be reproduced by a one-phonon thermal mixing model.”

The two-phonon thermal mixing model enables to reproduce, with a same set of parameters, both $\Gamma_S(T)$ and $\Gamma_L(T)$. In the appended figure, we display simulations obtained for the two configurations of bright-dark levels ordering. This figure shows that the temperature dependence of $\Gamma_S(T)$ and $\Gamma_L(T)$ is weakly sensitive to this level ordering, because the energy of the LO phonons involved in the thermalization (~ 3.5

meV) is one order of magnitude larger than the bright-dark splitting (0.3 meV). Hence, we cannot conclude on the bright-dark level ordering based on the temperature-dependent PL decay.

This figure has been added to the Supplementary file (Fig. S11) and we refer to it in the revised manuscript (page 16): “It is worth noting that when we reverse the bright-dark level ordering in the model, the calculated temperature dependences of $\Gamma_S(T)$ and $\Gamma_L(T)$ remain very similar and reproduce as well the experimental data (see the Supplementary Figure S11). This is due to the fact that the energy of the LO phonons involved in the thermalization of these levels is much larger than their splitting. Hence, no conclusion can be derived on the strength of the Rashba effect in these materials, although a Rashba effect can be inferred from the non-centrosymmetry of the reported low-temperature phase of FAPbI₃.”

Fig. S11. Modeling the temperature dependence of the PL decay with the two-phonon thermal mixing model, taking the dark level below or above the bright one.

The decay rates of the short (blue points) and the long (red points) components are the same as in Fig. 4. The blue curve and the red curve are the same simulated evolutions as in Fig. 4c and d. They are obtained by taking the dark level below the bright, and the parameters $\Gamma_B = 0.53 \text{ ns}^{-1}$, $\Gamma_D = 0.015 \text{ ns}^{-1}$, $\gamma_0 = 0.16 \text{ ns}^{-1}$, $E_{LO1A} = 3.3 \text{ meV}$, $E_{LO1B} = 3.6 \text{ meV}$. For comparison, the black curves are the simulated evolutions obtained after reversing the order of bright and dark levels in the model.

3- “In page 12 authors explain that they have applied an external magnetic field up to 7 T « to unveil the dark state emission line » but they did not observe any new PL line. They observed only a line broadening and they concluded that dark exciton line is very closed to bright exciton (their splitting is in sub-meV range). However, authors do not comment about the changes observed in the PL decay under magnetic field that has to be also affected by « brightening effect » of the dark exciton”.

Response:

We thank the referee for this remark. Indeed, we found that the long-lived decay component shortens and gains weight under a magnetic field of 7 T (see the appended figure). This constitutes a signature of a magnetic brightening of the dark state in FAPbI₃ NCs.

We added this figure in the Supplementary information file, and added this sentence in the main text: (page 13): “Moreover, under magnetic fields, the PL decay shows a shortening of its long-lived component with an increase of its weight (See the Supplementary Fig. S8).”

Fig. S8. Magnetic field effect on the PL decay.

(a) Temperature evolution of the PL decay for a single FAPbI₃ NC. (b) Under a magnetic field the long time component shortens and gains weight. The magnetic brightening of the dark state is less pronounced at 20 K than at 11 K due to a stronger thermal mixing between the bright and dark states.

4- “Could the authors give a more quantitative description of the two-phonon thermal mixing process between bright and dark exciton states?. For example, is it possible to link γ_0 rate to the electron phonon coupling strength, phonon dispersion.... ? γ_0 is a spin flip rate. Can authors explain why this two-phonon process is more probable for LO1 than for LO2 and why this process can mix a dark and a bright state?”

Response:

A two-phonon scattering process was introduced by Tsitsishvili et al. (Phys. Rev. B **66** (2002) 161405(R)) to model the thermal mixing within a radiative doublet in strongly confined semiconductor quantum dots with asymmetrical shapes, while one-phonon processes are strongly suppressed by a phonon-bottleneck effect. This energy level configuration with respect to phonon energies is very similar to that of our FAPbI₃ NCs. The two-phonon process is more probable to be done with LO₁ phonons than with LO₂ phonons because the characteristic temperature of mixing is ~40 K, which corresponds to an activation energy of 3.5 meV, matching the LO₁ mode energy.

Concerning the derivation of the characteristic spin-flip rate γ_0 , we added the following discussion in the Supplementary file and refer to it in the main text, with reference to the work of Tsitsishvili et al.

Usually, for small energy splittings between fine structure sublevels, the interactions with acoustic phonons are the dominant relaxation processes. However, in these soft materials displaying a high level of disorder in the FA cation positions, the interaction of the excitons with acoustic phonons is drastically reduced. In contrast, quasi-elastic two-phonon relaxation processes, in which one LO phonon is absorbed and another is emitted, can lead to efficient interlevel thermalization. The temperature dependence of the experimental decay rates is well reproduced using a two LO-phonon process. This observation is consistent with a dominant Fröhlich interaction associated with low energy optical phonons.

Relaxation rates associated to two-phonon processes may be computed from the second order Fermi golden rule ^{1,2}:

$$1/\tau = \frac{2\pi}{\hbar} \sum_q \sum_k \left| \sum_s \frac{M_q^{is} M_k^{sf}}{E_i - E_s + \hbar\omega_q + i\hbar\Gamma_s/2} + \frac{M_k^{is} M_q^{sf}}{E_i - E_s - \hbar\omega_k + i\hbar\Gamma_s/2} \right|^2 N_q(N_k + 1) \delta(E_f - E_i - \hbar\omega_q + \hbar\omega_k)$$

where $N_{q(k)} = 1/[\exp(\hbar\omega_{q(k)}/k_B T) - 1]$ is the Bose-Einstein occupation number for the absorbed (emitted) phonon, $1/\Gamma_s$ is the lifetime of the intermediate state s , $M_{q(k)}^{ab}$ are the matrix elements, $E_{i,s,f}$ are the energy levels of the initial, intermediate and final states, respectively.

Matrix elements for the Fröhlich interaction with optical phonons $\hbar\omega_q$ are :

$$M_q^{ab} = i \sqrt{\frac{e^2 \hbar \omega_q}{2q^2 V \epsilon_0} \left(\frac{1}{\epsilon_\infty^r} - \frac{1}{\epsilon_s^r} \right)} \langle \Psi_a^{e(h)}(\mathbf{r}) | e^{i\mathbf{q}\cdot\mathbf{r}} | \Psi_b^{e(h)}(\mathbf{r}) \rangle$$

where $\epsilon_{\infty(s)}^r$ is the high frequency(static) dielectric constant and $\Psi_a^{e(h)}(\mathbf{r})$ and $\Psi_b^{e(h)}(\mathbf{r})$ are the electron(hole) wavefunctions in the initial, intermediate or final

states.

Since the bare electron/hole phonon interaction does not contain spin operators, the matrix elements of the Fröhlich interaction are non-zero only if spin mixing of the involved exciton states occurs. Spin mixing might be related either to the exchange interaction³ or to the Rashba effect that is in principle triggered by a loss of inversion symmetry and enhanced by the large spin-orbit coupling in Pb-based compounds⁴. Non-centrosymmetry was indeed reported in the early study of the low temperature crystallographic structure of FAPbI₃⁵ and was recently confirmed⁶. In order to estimate the characteristic two-phonon mixing rate γ_0 , an elaborate description of the electronic structure of these nanocrystals requires detailed information on their low-temperature crystallographic structures, spectroscopic properties, morphology and ligands binding at the surface. This is beyond the scope of the present work.

Can authors comment on the polarization properties of the dark state PL, will be it modified by this process?"

Response:

We do not understand this point. We think that the reviewer meant how the polarization properties of the dark state is modified by the two phonon thermalization process when it is brightened by a magnetic field. Without magnetic field, the dark state does not emit, regardless of temperature. However, if this state is brightened by a magnetic field, its polarization should be the same as that of the closely lying bright state to which it is strongly coupled.

We did not investigate the polarization properties of the PL spectra. Due to fast spectral diffusion of these NCs, the spectral components of the band-edge exciton fine structure are buried in the emission line.

5- *"Parameters obtained from fitting of PL decay curves for different temperatures should be comment in more detail."*

Response:

To reproduce the temperature dependence of the short and the long components of the decays, as well as their relative weights:

- We choose two optical phonons that match the LO₁ phonon sideband (observed

in the PL spectrum), red-shifted by ~ 3.5 meV with respect to ZPL. The energies of these phonons are compatible with the observed activation temperature.

- The dark-bright splitting is explored between 200-500 μeV , i.e. within the ZPL linewidth and in accordance with the experimental observation of fine structure splittings in CsPbI_3 NCs.
- The determination of Γ_D and Γ_B is robust since these parameters are highly constrained by the values of the long and the short decay rates in the limits of low and high temperature regimes.

Indeed, in the low temperature regime where $k_B T \ll E_{LO1i}$ ($i=A,B$), we have

$$\Gamma_S \rightarrow \Gamma_B \text{ and } \Gamma_L \rightarrow \Gamma_D .$$

In the high temperature regime where $k_B T \gg E_{LO1i}$ ($i=A,B$), we have

$$\Gamma_S \rightarrow 2\gamma_0 (k_B T)^2 / (E_{LO1A} E_{LO1B}) \text{ and } \Gamma_L \rightarrow (\Gamma_B + \Gamma_D) / 2.$$

- The parameter γ_0 is accurately adjusted to reproduce the evolutions with temperature of both Γ_S and Γ_L .

We have added the following paragraph in the revised manuscript (page 15):

“To reproduce the experimental temperature dependence of the short and the long decay components, as well as their relative weights, we chose two optical phonon energies within the LO_1 sideband (centered around 3.5 meV). Their difference is explored between 200-500 μeV to match the dark-bright energy splitting. The determination of Γ_D and Γ_B is robust since these parameters are highly constrained by the values of the long and the short decay rates in the limits of low and high temperature regimes. Indeed, for $k_B T \ll E_{LO1i}$ ($i=A,B$), we have $\Gamma_S \rightarrow \Gamma_B$ and $\Gamma_L \rightarrow \Gamma_D$, while for $k_B T \gg E_{LO1i}$ ($i=A,B$), we have $\Gamma_S \rightarrow 2\gamma_0 (k_B T)^2 / (E_{LO1A} E_{LO1B})$ and $\Gamma_L \rightarrow (\Gamma_B + \Gamma_D) / 2$. The parameter γ_0 is accurately adjusted to reproduce the evolutions with temperature of both Γ_S and Γ_L .”

Minor remarks.

1- *“X-ray diffraction and spectroscopic data of Figure 1 are obtained at room temperature. I suggest to the authors to clearly give this information in the figure caption or in the different panels of this figure. The same for low temperature data.”*

Response:

In the revised manuscript, the title of the Fig. 1 caption is now *“Room-temperature properties of FAPbI_3 perovskite NCs.”*

2- *“At the end of page 4, the bulk bandgap of FAPbI_3 is given to be equal to 1.47 eV, i.e., 230 meV lower in energy than the emission of NCs. Authors explain that as a weak quantum size-effect in the studied NCs that have a size 2—3 times larger than the exciton Bohr radius. Another possible explanation is that measuring of the PL spectra very often does experimental determination of bandgap. In bulk materials PL spectra is some times dominated by bound excitons and not free excitons and bound excitons appear at lower energy than free excitons.”*

Response:

We agree with the reviewer on this point. In the revised manuscript, we suppressed the sentence where we compared the emission energy of the FAPbI₃ NCs to the reported bandgap energy of bulk FAPbI₃ deduced from PL measurements.

3- *“In Figure 3 a) authors give in blue the fitted spectra by using a triple Gaussian or Lorentzian profiles. From 55K until 150k it will be interesting to have in this figure the three Lorentzian or Gaussian curves used to fit as well as the whole fitted spectra. The same for similar figures in the supplementary information.”*

Response:

In the revised manuscript and the Supplementary file, we added multiple Lorentzian or Gaussian curves to fit these spectra, and justified the use of Lorentzian or Gaussian fits (see the response to the first point raised by the first reviewer).

4- *“In figure 4 a) it will be suitable to give the experimental data in a log-log representation in which the bi-exponential behaviour will be more evident. It will be interesting also to have the fitting curve superimposed on the experimental data.”*

Response:

As suggested by the reviewer, we added the fitting curves superimposed on the experimental decays in the revised manuscript. The log-log representation of the decays is displayed below for different temperatures. To our opinion, this representation is less adapted than the semi-log representation to show evidence for the evolution of the long component. We choose to keep the semi-log representation in the main text and added the appended figure in the Supplementary file (Supplementary Fig. 12).

Fig. S12. Temperature dependence of the PL decay of a single FAPbI₃ NC in log-log scale. The PL decay and fitting curves (red curves) are the same as those presented in Fig. 4a.

5- “In figure 4 d) could authors give the errors bars for Γ_L ?”

Response:

Actually the error bars are included in the initial version of the manuscript. For most of the points the error bar is smaller than the point size. In fact the long component decay rate Γ_L is accurately determined from the fits above 14 K.

6- “Authors should also add two references directly related with topics addressed in this work: fine structure- Ramade J. et al. *Nanoscale* 10, 6393 (2018), and dynamics of exciton PL in ensembles of perovskite NCs ; Chen L. *Nano Letters*. 18, 2071, (2018).”

Response:

These two references have been added to the revised manuscript.

References

1. Tsitsishvili, E., Baltz, R. V. & Kalt, H. Temperature dependence of polarization relaxation in semiconductor quantum dots. *Phys. Rev. B* **66**, 87 (2002).
2. Tsitsishvili, E. Light-hole exciton spin relaxation in quantum dots. *Phys. Rev. B* **91**, 178 (2015).
3. Nahálková, P. *et al.* Two-phonon assisted exciton spin relaxation due to exchange interaction in spherical quantum dots. *Phys. Rev. B* **75**, 1031 (2007).
4. Becker, M. A. *et al.* Bright triplet excitons in caesium lead halide perovskites. *Nature* **553**, 189–193 (2018).
5. Stoumpos, C. C. *et al.* Crystal Growth of the Perovskite Semiconductor CsPbBr₃: A New Material for High-Energy Radiation Detection. *Cryst. Growth Des.* **13**, 2722–2727 (2013).
6. Weber, O. J. *et al.* Phase Behavior and Polymorphism of Formamidinium Lead Iodide. *Chemistry of Materials* **30**, 3768–3778 (2018).

REVIEWERS' COMMENTS:

Reviewer #1 (Remarks to the Author):

I would like to thank the authors to improve the manuscript by adding a new piece of experimental data and additional analyses. The revised manuscript now gives more clear evaluation on the exciton fine structure and exciton-phonon coupling and addresses my concern and comments. Somehow, I still wonder whether the findings in this manuscript can be applicable to other types of perovskites NCs. But, for now, I recommend that this manuscript be published in Nature Communication.

Reviewer #2 (Remarks to the Author):

Report on the paper entitled "Unravelling exciton-phonon coupling in individual FAPbI₃ nanocrystals emitting near-infrared single photons" by M. Fu et al.

The authors answered referee's questions with a lot of honesty and they improved the explanations and arguments of his work according to the referee's requests. Then, I recommended the publication of this new version of the manuscript in Nature Communications.